# *Bacillus subtilis*: As an Efficient Bacterial Strain for the Reclamation of Water Loaded with Textile Azo Dye, Orange II

**DOI:** 10.3390/ijms231810637

**Published:** 2022-09-13

**Authors:** Muhammad Ikram, Mohammad Naeem, Muhammad Zahoor, Marlia Mohd Hanafiah, Adeleke Abdulrahman Oyekanmi, Noor Ul Islam, Midrar Ullah, Mater H. Mahnashi, Amer Al Ali, Naif A. Jalal, Farkad Bantun, Aiman M. Momenah, Abdul Sadiq

**Affiliations:** 1Department of Chemistry, Abdul Wali Khan University Mardan, Mardan District, Mardan 23200, Khyber Pakhtunkhwa, Pakistan; 2Department of Biochemistry, University of Malakand, Dir Lower, Chakdara 18800, Khyber Pakhtunkhwa, Pakistan; 3Department of Earth Sciences and Environment, Faculty of Science and Technology, Universiti Kebangsaan Malaysia, Bangi 43600, Selangor Darul Ehsan, Malaysia; 4Centre for Tropical Climate Change System, Institute of Climate Change, Universiti Kebangsaan Malaysia, Bangi 43600, Selangor, Malaysia; 5Department of Chemistry, University of Malakand, Dir Lower, Chakdara 18800, Khyber Pakhtunkhwa, Pakistan; 6Department of Biotechnology, Shaheed Benazir Bhutto University, Dir Upper, Sheringal 00384, Khyber Pakhtunkhwa, Pakistan; 7Department of Pharmaceutical Chemistry, College of Pharmacy, Najran University, Najran 66462, Saudi Arabia; 8Department of Clinical Laboratory Sciences, Faculty of Applied Medical Sciences, University of Bisha, 255, Al Nakhil, Bisha 67714, Saudi Arabia; 9Department of Microbiology, Faculty of Medicine, Umm Al-Qura University, Makkah 24382, Saudi Arabia; 10Department of Pharmacy, Faculty of Biological Sciences, University of Malakand, Dir Lower, Chakdara 18800, Khyber Pakhtunkhwa, Pakistan

**Keywords:** bacterial strains, degradation, metabolites, toxicity, wastewater, water pollution

## Abstract

The azo dye orange II is used extensively in the textile sector for coloring fabrics. High concentrations of it are released into aqueous environments through textile effluents. Therefore, its removal from textile wastewater and effluents is necessary. Herein, initially, we tested 11 bacterial strains for their capabilities in the degradation of orange II dye. It was revealed in the preliminary data that *B. subtilis* can more potently degrade the selected dye, which was thus used in the subsequent experiments. To achieve maximum decolorization, the experimental conditions were optimized whereby maximum degradation was achieved at: a 25 ppm dye concentration, pH 7, a temperature of 35 °C, a 1000 mg/L concentration of glucose, a 1000 mg/L urea concentration, a 666.66 mg/L NaCl concentration, an incubation period of 3 days, and with hydroquinone as a redox mediator at a concentration of 66.66 mg/L. The effects of the interaction of the operational factors were further confirmed using response surface methodology, which revealed that at optimum conditions of pH 6.45, a dye concentration of 17.07 mg/L, and an incubation time of 9.96 h at 45.38 °C, the maximum degradation of orange II can be obtained at a desirability coefficient of 1, estimated using the central composite design (CCD). To understand the underlying principles of degradation of the metabolites in the aliquot mixture at the optimized condition, the study steps were extracted and analyzed using GC-MS(Gas Chromatography Mass Spectrometry), FTIR(Fourier Transform Infrared Spectroscopy), ^1^H and carbon 13 NMR(Nuclear Magnetic Resonance Spectroscopy). The GC-MS pattern revealed that the original dye was degraded into o-xylene and naphthalene. Naphthalene was even obtained in a pure state through silica gel column isolation and confirmed using ^1^H and ^13^C NMR spectroscopic analysis. Phytotoxicity tests on *Vigna radiata* were also conducted and the results confirmed that the dye metabolites were less toxic than the parent dye. These results emphasize that *B. subtilis* should be used as a potential strain for the bioremediation of textile effluents containing orange II and other toxic azo dyes.

## 1. Introduction

Synthetic dyes including azo, heterocyclic, triphenylmethane and anthraquinone dyes are used extensively in the textile, paper, printing, food and cosmetic industries. Annually, worldwide, approximately 2.8 × 10^5^ tons of textile wastewater loaded with dyes is discharged into the environment [1]. These dyes are extremely stable and, under light and aerobic conditions, shows resistance to degradation. Most of the dyes and their transformation products are mutagenic and carcinogenic in nature [2]. In the textile industry, for dyeing purposes, dyes containing about 80% azo are used. Approximately 10–15% out of these dyes do not bind to the fibers during the dyeing process. Consequently, these dyes are released to the environment [3,4]. The effluents of the tanning and textile industries accumulate in the environment, causing ecosystem damage and phytotoxicity [5]. High concentrations of organic pollutants are hazardous to the aquatic environment and affect the health of the flora and fauna. Therefore, the discharge of effluents without sufficient treatment has a substantial negative impact on the environment [6]. Some of the dyes, as well as their transformation metabolites, are reported as toxic, carcinogenic and mutagenic. Incomplete biodegradation under anaerobic conditions produces aromatic amines that have major toxic effects on the living biota in the environment [7]. Orange II dye, also known as acid orange 7, is also widely used in textile industries due to its characteristic properties. Orange II (acid orange 7) poses health risks to humans, just like other azo dyes that are produced in industrial wastewater. Orange II (acid orange 7) is extremely hazardous and has the possibility of irritating the eyes, skin, mucous membranes, eyes and upper respiratory tract, as well as causing severe headache, dizziness, nausea and loss of bone marrow; this can result in anemia. The ingestion of it can potentially result in death because it is carcinogenic in nature and can cause tumor formation. Orange II is quickly reduced and broken down by enzymes inside the human body where it can also produce aromatic amines that can lead to methemoglobinemia [8]. Dyes containing effluents have harmful impacts on water, soil, air, plants and human beings. They cause several diseases in human beings such as allergy, asthma, eye and skin irritation including dermatitis, nausea, cancer, tuberculosis, gene mutations, hemorrhage and heart diseases [9]. Synthetic dyes are produced on a large scale and used widely, which severely contaminates surface and groundwater sources near dye industries. This puts more demand on water resources that are clean and increases the problem of water quality, thereby causing scarcity of portable water for domestic utilization [10]. Water contaminated with dyes is not fit for domestic purposes and drinking due to its hazardous effect on life forms [11]. Furthermore, the negative impacts of azo dyes are faced by surface water as well as underground water. These effects can lead to the blockage of light transmission and can affect the algal photosynthesis pathway, as well as other aquatic plants that would cause a reduction in dissolved oxygen (DO) in water bodies for living organisms [12]. Due to poisonous, mutagenic and carcinogenic chemicals, the majority of industrial effluents are hazardous to health. As a result, industrial waste effluents usually need to be treated before being reused or released into open water bodies [13,14]. 

Various physico-chemical methods including adsorption, precipitation, coagulation, photodegradation and chemical oxidation processes can be used for the removal and decolorization of industrial dyes. However, due to high cost, intensive energy requirements and low efficiency, these methods have not been widely applied for better results. Many new approaches have been employed for dye decolorization. One of these potential approaches is microbial decolorization and biodegradation. The biodegradation of dyes is considered to be an ecofriendly and cost-effective technique [15]. *Rhodococci* are viewed as potential biodegraders and biotransformers of several xenobiotics in recent years [16]. The use of microorganisms for the purpose of dye degradation has various advantages. The dyes are degraded completely in a cost-effective manner, requiring less water and producing less sludge. Moreover, the microbiological treatment of dyes is an efficient, ecofriendly and sustainable technique [17,18]. In dye biodegradation, microbes are used which are indigenous in their contaminated habitats. These microbes include bacteria, algae and fungi. Among these organisms, bacteria are preferred because they grow rapidly under different conditions in varied environments [19,20]. Several studies have been reported in the literature on azo dye’s degradation under anaerobic and aerobic conditions using bacterial strains. According to Pharm et al. [21] the use of microbes for the dye’s degradation in recent years has been reported in the literature in a wide range of studies. The synergistic metabolic pathways of microbes have played a vital role in the efficient degradation of dyes. In the literature, biodegradation of the dyes has been extensively reported utilizing a variety of aerobic and anaerobic bacterial strains, including *Clostridium* sp., *Staphylococcus hominis, Micrococcus luteus, Micrococcus glutamicus, Sphingomonas melonis, Anoxybacillus* sp., *Brevibacillus laterosporus, Clostridium bifermentans, Galactomyces* sp., *Pseudomonas putida, Bacillus cohnii, Brevibacterium* sp., *Aeromonas hydrophila, Providencia* sp., *Staphylococcus arlettae, Marinobacter* sp., *Sphingobacterium* sp., *Aeromonas hydrophila, Acinetobacter calcoaceticusi, Enterococcus casseliflavus, Acinetobacter baumannii, Pseudomonas aeruginosa, Acinetobacter junii, Brevibacterium* sp., *Staphylococcus aureus, Providencia* sp., *Shewanella* sp., *Alishewanella* sp., *Rhizobium* sp., *Bacillus cereus, Proteus mirabilis, Pseudomonas luteola, Pseudomonas aeruginosa* and *Morganella* sp. [22,23,24,25]. Sananraj et al. reported that *B. subtilis* is the best decolorizer with 99.05% decolorization potential for Direct Green-PLS azo dye. The *B. subtilis* strain was isolated from textile industrial effluent in a reported study [26]. Dye degradation occurs through the action of different enzymes such as azo reductases, laccases, veratryl alcohol oxidase, tyrosinases and peroxidases. The most common enzyme of the bacterial system is azoreductase [5]. The biodegradation of dyes using bacterial strains and the isolation of metabolites has not been given considerable attention in the literature. In this study, *B. subtilis*, a Gram-positive bacterium, was chosen because it can readily produce enzymes such as azoreductase at normal temperatures up to 37 °C [27]. Due to azoreductase activity, *Bacillus* sp. degrades azo dyes and transforms them into colorless compounds. The azoreductase enzyme gene has been found in *B. subtilis* [28]. The degradation kinetics of another azo dye, p-aminoazobenzene, by *B. subtilis* has been reported in the literature. The results prove that *B. subtilis* could break the azo linkage of the dye in the presence of glucose (carbon source), resulting in the formation of p-phenylenediamine and aniline products [29]. Recently, Ullah et al. reported an insight into the mechanism of biodegradation of another azo dye, brown 706, by *Pseudomonas aeruginosa* sand provided an insight into the degradation mechanism. The *Bacillus* species has a good decolorizing potential for Orange 3R, according to a study reported in the literature [30]. The degradation pathway, degradation mechanism and metabolite isolation of orange II dye by *B. subtilis* have not been utilized properly in any literature studies. Therefore, the present study was conducted to investigate its degradation efficiency for orange II (acid orange 7) dye. 

As mentioned above, in the literature, very few studies on the nature of the resultant metabolites of the degradation mechanism have been published. Although it has been found that bacterial strains are capable of degrading dyes, it is of immense importance to know the nature of the metabolites formed. In some conditions, the original dye is less toxic while the formed metabolites are more toxic and, in such instances, the occurrence of transformations does more harm than the reclamation [31,32]. In such cases, the effects of physicochemical parameters such as dye concentration, agitation and oxygen level, pH, carbon and nitrogen sources are evaluated. These parameters would help in limiting hazardous product formation under a given set of conditions [33].

The present research aimed to investigate a potent bacterial strain for orange II (acid orange 7) degradation out of the available strains and to understand the degradation pathway of the selected strain. The effects of various physicochemical parameters were also determined in order to obtain the optimum orange II (acid orange 7) degradation conditions in a one-at-a-time optimization procedure and the interactive effect of process conditions using a response surface methodology (RSM) central composite design (CCD). The extracted isolates were then subjected to column chromatography to obtain metabolites in a pure form. The isolated metabolites were then subjected to structural elucidation using spectroscopic techniques such as FTIR, NMR and GCMS. It is worth mentioning here that some of the strains used in our study belong to normal flora, while most of these bacterial strains are opportunistic; therefore, precautionary measures and safety are needed for researchers while conducting biodegradation of dyes using these bacterial strains.

## 2. Results and Discussion

### 2.1. Highly Potent Bacterium for Orange II Degradation

Different bacterial strains have different decolorization potential. In our study, 11 bacterial strains were used; out of these bacterial strains, *B. subtilis* emerges as the most potent strain for orange II biodegradation. The percentage decolorization of this bacterium was 85.75%. The percentage decolorization potential of all other bacterial strains is also given in Figure 1.

### 2.2. Optimal Conditions of Orange II Degradation 

#### 2.2.1. Impact of Dye Concentration on Orange II Degradation 

In Figure 2a, the effect of the concentration of orange II on the biodegradation rate is shown. It is clear from the percentage degradation values that highest degradation rate (78.96%) was observed for the 25 ppm dye solution, which was thus considered nontoxic for *B. subtilis* since a higher concentration of dye is toxic and decreases the bacterial dye degradation activity. Zhuang et al. [34] reported that bacterial activity and degradation potential are inhibited due to toxicity and the blockage of enzyme active sites with high concentrations of dye. Sunar et al. [35] claimed, in the literature, that an increase in dye concentration may cause toxicity, the inhibition of bacterial metabolic activity, saturation of the bacteria cells with dye, or blockage of the active sites of the azo reductase. The average dye content in wastewater from the textile sector is between 16 and 50 mg/L [36]. Additionally, a higher concentration of dye needs more biomass of bacteria [37]. From the above discussion, it can be stated that rate of decolorization (degradation) decreases with increasing dye concentrations. In Appendix A, the percentage degradation of orange II at different concentrations is given. 

#### 2.2.2. pH Effect on Orange II Degradation

Enzymatic activity and bacterial biodegradation potential are affected by the pH of the solution [38]. For better decolorization and degradation optima pH is necessary. In Figure 2b, the impact of pH on orange II degradation is shown. When pH increases from acidic to basic, an increase in decolorization occurs. Maximum decolorization is shown at pH 7, which is 62.09%; this depicts that extreme alkaline and acidic conditions affect the growth and enzymatic activity of bacteria. According to Solis et al. [39], the optimal pH range is 6 to 10 for decolorization of the of dyes by bacteria, while Agrawal et al. [40] reported a preference for alkaline conditions for bacterial decolorization. Optimal pH ranges of 6 to 10 are utilized in most of the textile industrial processes. Extreme acidic or alkaline conditions decrease the degradation rate [41]. In the literature, most of the azo dyes reducing bacterial strains that have been reported so far have been able to reduce the azo dye at a pH near 7 [42,43,44]. This could be connected to dye molecules’ ionic states, as in a neutral state, they can be transported through cell membranes by the bacterial system easily [45]. Above pH 7, decolorization by bacteria was seen to decrease as a result of a reduction in the azo bond and the formation of aromatic amines, which are more alkaline than the parent azo dye [46,47]. Recently, Ullah et al. [41] have claimed the biological degradation of another azo dye, methyl orange, by *Pseudomonas aeruginosa* at pH 7. Appendix A shows the percentage degradation of orange II at different pH levels.

#### 2.2.3. Impact of Incubation Duration (Time) on Orange II Degradation

In Figure 2c the time effect on the degradation of orange II by *B. subtilis* is shown. The dye percentage degradation was recorded up to 21 days continuously. On the third day of incubation, a maximum decolorization of 54.49% was observed. The degradation rate then remained steady until day 21. In the initial stage, bacterial biomass production was high, but with the passage of time, it decreased because the bacterial colonies compete for nutrient availability. Our results are closely related with the literature findings of Ullah et al. [41] who performed biological mineralization of another azo dye, methyl orange, by *Pseudomonas aeruginosa*, wherein a 72 h experimental time was found to be optimum. In another literature finding, the biodegradation of the azo dye brown 706 by *P. aeruginosa* was achieved after 3 days of incubation time. Appendix A indicates the percentage degradation of dye at different time intervals.

#### 2.2.4. Temperature Effect on Orange II Dye Degradation

Temperature affects the dye biodegradation potential of bacteria. Figure 2d shows the impact of temperature on orange II degradation. An optimum temperature is needed for bacterial growth; above or below this temperature, bacteria cannot grow and their survival is then difficult, consequently affecting the degradation potential of bacteria. *B. subtilis* showed highest decolorization rate (63.34%) at 35 °C, which means that above and below this temperature, its decolorization rate decreases as its growth rate becomes slow. Dye degradation is a metabolic process so changing the temperature from the optimum range results in a decrease in the decolorization rate of the dye. Due to high temperature, proteins in the bacterial system become inactive and cell structures such as membranes are probably affected [48]. A similar finding has been reported in a study where the decolorization of another azo dye, Reactive Red M8B, by *B. subtilis* was achieved at 35 °C [49]. It was stablished that the decolorization/degradation of the reactive azo dyes was achieved at a range of temperatures from 35 to 45 °C. Above the optimum temperature, a decrease in the decolorization occurs due to the denaturation of bacterial enzymes [50]. It has been reported in the literature that in microbial physiology, changing temperature suddenly alters the activation energy [51]. According to Anjaneya et al. [52], the inactivation of bacterial enzymes occurs at high temperatures, which results in a decrease in the dye decolorization rate of bacteria. The percentage degradation at different temperatures is given in Appendix A.

#### 2.2.5. Impact of Glucose on Orange II Degradation

Glucose serves as a source of carbon for bacteria; this is why for the production of bacterial biomass and dye break down, a sufficient glucose amount is necessary. As some of the dyes are complex and difficult to degrade, an additional carbon source is required to provide carbon [53]. According to reports, the most efficient and convenient carbon source for the microbial degradation of dyes or dye intermediates is glucose [54]. In Figure 2e the impact of glucose concentration on dye decolorization (degradation) is shown. The highest degradation rate (71.66%) was shown at 1000 mg/L glucose supplementation. However, the highest rate of degradation was also seen with low glucose supplementation as opposed to high amounts, demonstrating that glucose supplementation is less effective at promoting decolorization activity. This is likely because bacterial cells prefer to assimilate the additional carbon sources rather than using dye molecules as the carbon source [55]. The degradation activity further decreases as the concentration of glucose increases; this may be due to the negative effect on sugar catabolic repression of the bacterial metabolic pathway [56]. In Appendix A, the percentage degradation of orange II dye at different concentrations of glucose has been represented.

#### 2.2.6. Impact of Urea on Orange II Dye Degradation

Urea is used as a nitrogen source by bacteria, so bacteria need a sufficient amount of urea to degrade the given dye. According to Chang et al. [57] organic nitrogen supplements such as urea can regenerate NADH, which acts as an electron donor for the reduction of dye molecule by microbes to ensure effective biodegradation. The influence of urea concentration on the percentage degradation of orange II dye is given in Figure 2f. A high degradation rate was recorded at a 1000 mg/L urea concentration. Ikram et al. reported that a high concentration of urea is toxic for bacteria. The highest degradation was achieved using a low concentration of urea, and by further increasing the concentration of urea, the degradation potential of *Escherichia coli* was decreased. The dye break-down activity decreased at high concentrations of urea, which is due to the induced toxicity of urea [58]. In Appendix A the percentage degradation of orange II at different concentrations of urea supplementation has been given.

#### 2.2.7. Impact of Sodium Chloride Concentration on Degradation

Figure 2g indicates sodium chloride concentration’s effect on orange II dye degradation. The bacterial biodegradation ability decreases as the concentration of salt rises. An optimal salt concentration is required for efficient degradation. The optimal salt concentration in our experiment was 666.66 mg/L as the highest percentage degradation (58.69%) was recorded at this concentration. Recently Asad et al. also reported the degradation of the azo dye brown 706 using *Pseudomonas aeruginosa* in their experiment at a low concentration of salt (0.1 gm/15 mL) [59]. A high salt concentration results in plasmolysis of bacterial cells, and bacterial growth is reduced; consequently, its dye degradation potential decreases [60,61]. Appendix A depicts the percentage degradation of orange II at different concentrations of sodium chloride (NaCl).

#### 2.2.8. Redox Mediators and Their Effects on Selected Dye Degradation

Some chemical compounds that act as electron donors and acceptors are utilized by bacteria as redox mediators. The azo linkage of the dyes is reduced by bacterial strains either in aerobic or anaerobic conditions. Some bacterial enzymes such as azoreductase and oxidoreductase are involved in oxidation-reduction reactions so their activities are induced by redox mediators. The effect of redox mediators on dye biodegradation is shown in Figure 2h; hydroquinone showed the highest degradation rate (60.93%) for the given dye. The addition of a slight redox mediator is sufficient to reduce the steric hindrance of the dye molecules and speed up the electron transfer reaction [62,63]. According to Olukanni et al. [64] the presence of hydroquinone as a mediator increased the degradation yield to 94.41% after 5 h of incubation time compared to 79.35% in the control study when the dye was treated in nutrient broth without the addition of any redox mediator. These results are closely related and remarkably comparable to other reported studies in the literature where the use of redox mediators has shown the highest degradation potential. During the anerobic reduction of azo dye, the rate-limiting or slowing step has been shown to be the transfer of reducing equivalents from the co-substrate or primary electron donor to the azo dye (final electron acceptor) [65]. Appendix A shows orange II percentage degradation by *B. subtilis* using different Redox mediators.

### 2.3. Orange II Biodegradation at Optimum Conditions

Initially, the effects of various parameters on dye degradation by *B. subtilis* were determined. These parameters include dye concentration, sodium chloride concentration, urea concentration, sugar concentration, pH, time, temperature and redox mediators. These optimal conditions were then employed in a single final experiment. The optimum conditions determined were 25 ppm dye concentration, a temperature of 35 °C, a 1000 mg/L glucose concentration, pH 7, a 666.66 mg/L sodium chloride concentration, a 1000 mg/L urea concentration, 66.66 mg/L hydroquinone as a redox mediator, and a 3-day incubation duration. About 83.37% degradation/decolorization was achieved when these optimal physiochemical conditions were combined in one experiment. Figure 3a,b shows the orange II dye color before and after bacterial treatment under optimum conditions. There is significant decolorization, as well as visual changes, after bacterial treatment, which indicates that the dye has been degraded by the bacteria and some new products have been formed. 

### 2.4. Most Significant Parameters and Response Surface Optimization for Degradation of Orange II 

The effects of the process conditions of four significant parameters, which include, pH, incubation time, dye concentration and temperature, were designed according to Table 1. Optimization of the effects of the operational factors on orange II degradation was conducted using Design-Expert 6.0.4 software using the response surface of the second-order quadratic function for the prediction of percentage dye degradation, according to Equation (1).
(1)$$Y\;\beta_{0}+\sum_{I=1}^{K}\beta_{I}X_{i}\;\;\sum_{I=1}^{K}\beta_{II}X_{i}^{2}+⅀\sum_{I>j}^{K}\beta_{Ij}X_{1}X_{j}\;+\;€$$
where *β*_0_ and *β**_I_* represent constant coefficients, *X**_i_* indicates the linear coefficient of the input factors, *β**_II_* represents the quadratic coefficient of the input factor *X**_i_*, *β**_ij_* evaluates the interaction coefficient between the input parameters *X**_i_* and *X**_j_*, and *€* indicates the error of the model.

The significance of the interaction between process variables, as indicated in the relationship between the predicted and observed values, influenced orange II degradation, as illustrated in Table 2. This is reflected in the linear relationship as indicated in the percentage probability plot, and the relationship between the predicted and observed values as shown in Figure 4a,b, respectively. The evaluation of the second-order quadratic model expressing the degradation of orange II is represented in Equation (2).
(2)$$Y=83.40-0.72X_{1}+1.74X_{2}–0.92X_{3}-1.20X_{4}–16.21X_{1}{}^{2}-2.69X_{2}{}^{2}-2.75X_{3}{}^{2}-1.95X_{4}{}^{2}-1.58X_{1}X_{2}-0.26X_{1}X_{3}+0.91X_{1}X_{4}+4.10X_{2}X_{3}-2.1X_{2}X_{4}+1.05X_{2}X_{4}-1.09X_{3}X_{4}$$

The suitability of the of the model for the analysis of the degradation efficiency of the bacteria strain was well described based on the result of the analysis of variance (ANOVA), as illustrated in Table 3. The high correlation coefficient R^2^ and a close relationship with the values of R^2^ ‘adjust’ were observed as 0.8950 and 0.7969, respectively. This was achieved with a low probability coefficient (*p* < 0.05) and high F-value, indicating that the operation conditions have a profound effect on the biodegrading of orange II using *B. subtilis*. The value of adequate (Adeq) precision was found to be 12.233 which indicated that the signal-to-noise ratio did not affect the prediction of process variables since a model is desirable if the Adeq precision is greater than 4 [66]. The interaction of the operational factors revealed that the effect of combined parameters influenced the biodegradation efficiency of the bacterial strain. This is indicated in Figure 5a–d. The interactions between the process variables resulted in increased degradation efficiency of *B. subtilis* for orange II biodegradation. This was achieved under optimum conditions of pH 6.45, a dye concentration of 17.07 mg/L and an incubation time of 9.96 h at 45.38 °C, and a desirability coefficient of 1 was obtained, which indicated that orange II degradation was well predicted by the model under the operational conditions investigated.

### 2.5. Characterization Study of Orange II Biodegradation Products

The characterization of metabolites of the biodegradation of orange II dye was analyzed using Fourier Transform Infrared (FT-IR), Gas Chromatography–Mass Spectrometry (GC-MS) and Nuclear Magnetic Resonance (NMR) Spectroscopy.

#### 2.5.1. Fourier Transform Infrared Spectroscopic Analysis

In Figure 6a, the FTIR spectra of undegraded dye is shown. The peak at 3400.77 cm^−1^ is attributed to amine N–H stretching, while the peak at 1619.27 cm^−1^ represents N=N stretching; the peak at 1597.72 cm^−1^ represents C=C aromatic stretching; the peak at 1451.44 cm^−1^ represents aromatic ring C–H stretching; the peaks at 1181.41 cm^−1^ and 1118.86 cm^−1^ correspond to the benzene ring attached to the C–N group; and the peak at 1032.89 cm^−1^ represents S=O stretching attached to benzene ring. Figure 6b shows the FTIR spectra of degraded dye. The peaks at 2960.19 cm^−1^ and 2929.55 cm^−1^ represent C–H stretching, while the peak at 1641.46 cm^−1^ corresponds to C=C stretching of the aromatic ring; the peak at 1716.49 cm^−1^ represents C=O stretching. The peaks at 1102.89 cm^−1^ and 1268.72 cm^−1^ represent C–O stretching; and the peak at 731.15 cm^−1^ represents C=C stretching of the aromatic ring. The peak at 1619.27 cm^−1^ corresponding to N=N stretching has disappeared, which clearly indicates a reduction in the azo bond due to the bacterial azoreductase enzyme. When the FTIR spectra of the original orange II dye reduces to the spectra of its degraded form, significant changes can be seen in their peaks. Some peaks have vanished while some new peaks have been formed, which indicates that the orange II dye has been degraded and metabolites or new compounds have been formed.

#### 2.5.2. Gas Chromatography–Mass Spectrometry (GC-MS) Analysis

The GC and GC-MS chromatograms of the degraded dye products or metabolites are given in Figure 7 and Figure 8a,b, respectively. The orange II metabolites found at an RT of 6.99 min and an RT of 2.24 min, with charge-to-mass values of 128 and 106 m/z that closely relate to the orange II structure, were identified as naphthalene and o-xylene, as shown in Table 4. These two new compounds/metabolites were formed as a result of the breakdown of the azo dye linkage across both sides which is also clear from the FTIR spectra. Subsequent enzymatic action of the bacterial system transformed the orange II dye to naphthalene and ortho-xylene. Even the naphthalene was isolated in a pure state during isolation from the silica gel column, and its structure was also further confirmed using ^1^HNMR and ^13^C NMR analysis. 

#### 2.5.3. NMR Spectroscopic Analysis

The metabolites extracted in ethyl acetate solvent were then passed through the silica gel column for purification and isolation. The fractions or metabolites collected from the silica gel column after elution by different solvents were then analyzed using Proton (^1^H) and carbon 13(^13^C) NMR spectroscopy. The ^1^H NMR and carbon 13 NMR spectra of the original orange II are shown in Figure 9 and Figure 10, respectively. By comparing the ^1^H and carbon 13 NMR spectra of the original orange II dye to those of the metabolites formed through biodegradation by *B. subtilis*, significant changes were found. Out of these metabolites, only naphthalene was obtained in a pure state in the silica gel column fractions. The ^1^H and carbon 13 NMR spectra of the isolated fraction are shown in Figure 11 and Figure 12, respectively.

In Figure 11 the ^1^H-NMR analysis shows that the signal appears as two peaks at δ 7.88 and δ 7.49 ppm, with both integrated for the four H atoms. The peak at δ 39.0 ppm is due to the methyl in DMSO. Figure 12 shows three peaks, indicating the signals from three types of carbon atom on an aromatic ring. These peaks are at δ 126.42, δ 128.19 and δ 133.46 ppm, respectively. All these peaks are related to ^13^C on sp^2^ CH groups. Therefore, if the ^1^H-NMR and ^13^C-NMR analysis results in Figure 11 and Figure 12 are combined together, a chemical compound with the molecular formula C_10_H_8_ (naphthalene) is formed. The chemical structure of the isolated metabolite (naphthalene) is given below in Figure 13.

### 2.6. Phytotoxicity Analysis

To assess the phytotoxicity of chemicals and industrial wastewaters, short-term and sensitive ecotoxicological bioassays such as seed germination and early seedling growth tests are normally utilized worldwide [67]. The seeds of *Vigna radiata* showed better % germination in treated dye water compared to untreated dye water. The results also revealed that that shoot and root length in treated dye water was more than in untreated dye, which shows the less-toxic nature of the dye metabolites formed; this is clear from Table 5. The percentage germination, shoot and root length in tap water, and untreated and treated orange II dye are shown in Figure 14, Figure 15 and Figure 16, respectively. Barathi et al. also reported that the metabolites produced by the degradation of the azo dye Reactive blue 160 by *B. subtilis* were less toxic compared to the parent dye. They conducted phytotoxicity experiments on green gram (*Vigna radiata) and corn*. In the case of treated dye water used for the plants, the % germination and shoot and root length were higher than those of the plant grown in untreated dye water [68].

### 2.7. Proposed Mechanism for Orange II Degradation by B. subtilis

Bacteria contain enzymes such as oxidoreductase, veratryl alcohol oxidase, azoreductase, peroxidase and laccase, which allows them to break down dyes in wastewater [69,70]. A wide range of bacteria have azoreductases, such as *Enterococcus faecalis YZ 66, Pseudomonas* sp. and *Bacillus* sp. [71,72,73]. Aerobic bacteria have oxidoreductive enzymes that can symmetrically or asymmetrically split dye molecules. They could also cause deamination and desulfonation. As a result, aerobic bacteria can break down different dye structures [74]. The enzyme azoreductase present in bacteria breaks the azo bond (–N=N– bond) in both anaerobic and aerobic conditions. Through the action of the azoreductase enzyme, the azo linkage is broken and reduces orange 2 dye, consequently, to two substituted benzene derivatives, which are 1-amino-2-napthol and sodium 4-aminobenzenesulfonate. Through the deamination process, the deaminase enzyme of bacteria could convert 1-amino-2-naphthol to naphthol. At last, the dexydroxylase enzyme of bacterial system converts naphthol to naphthalene. The other part of the dye molecule, sodium 4-aminobenzenesulfonate, through desulfonation, is converted into aniline. Afterwards, the aniline is converted to benzene via the deamination process through the deaminase enzyme. The methylation of benzene forms o-xylene in the last step of the reaction. In a bacterial system, the methyl donors; methyl tetrahydrofolate and S-adenosyl-methionine are already present and are claimed to ensure the methylation of benzene [75]. Figure 17 shows the proposed and possible mechanisms for the orange II degradation by the *B. subtilis* strain.

## 3. Materials and Methods

### 3.1. Dye and Other Reagents

The orange II dye was provided by the textile mill located in Sindh, Pakistan. All other chemicals including ethyl acetate, n-hexane, hydrochloric acid, nutrient broth, sodium chloride, urea, glucose, sodium hydroxide and redox mediators (uric acid, sodium benzoate, hydroquinone and EDTA) were of analytical grade and had a purity level. The other chemical reagents used in the study were purchased from Sigma Aldrich, Darmstadt, Germany. In Figure 18, the orange II dye structure is shown [76].

### 3.2. Bacteria Cultures

*Staphylococcus aureus* (ATCC 27700), *Staphylococcus epidermidis* (ATCC 14990), *Escherichia coli* (ATCC25922), *Citrobacter amalonaticus* (ATCC25405), *B. subtilis* (ATCC6633), *Xanthomonas campestris* (ATCC-13951), *Proteus mirabilis* (ATCC-29906), *Enterobacter sakazakii* (ATCC 29544), *Pseudomonas aeruginosa* (ATCC27853), *Salmonella enterica* (ATCC43971) and *Streptococcus pyogenes* (ATCC12344) were used to evaluate the biodegradation capabilities. These cultures were provided by the Biotechnology Department of the University of Malakand and the Microbiology Department of Abdul Wali Khan University Mardan, Khyber Pakhtunkhwa, Pakistan. 

### 3.3. Preparation of Dye Stock Solution and Growth Media (Nutrient Broth)

Using digital balance, orange II dye in an amount of 0.04 g was accurately weighed and dissolved in a little amount of distilled water in a conical flask. After complete dissolution with shaking for 5 min, the volume of the solution was raised to 1000 mL to achieve the desired concentration solution of 0.04 g/L (40 ppm). This stock solution was used in the subsequent experiments. Nutrient broth was used as growth media for the bacterial cultures. Nutrient broth powder in a specified amount was added into distilled water (13 g in 1000 mL of distilled water). To kill unwanted microbes and avoid the contamination of other bacteria, the nutrient broth, test tubes, conical flasks and other glassware were completely sterilized (at 121 °C) in an autoclave (A55 Autoclave Robust Technology Zhubei Taiwan) for 3 h. 

### 3.4. Culture Growth

The test tubes and nutrient broth solution were moved from the autoclave to the laminar flow hood. The test tubes were labeled before the inoculation of the bacterial strain. To each test tube, we added 10 mL nutrient broth solution. After culture inoculation, to ensure the growth of the selected bacteria, the test tubes were kept in an incubator (Laboratory Hot Air Oven, DIGISystem Laboratory Instruments Corporation) at 37 °C for 24 h.

### 3.5. Degradation/Decolorization Experiments

After 24 h of incubation where bacteria flourished in each tube, 5 mL of the dye stock solution was introduced to each test tube. The absorbance of the extracted supernatant was measured before decolorization or degradation. After three days, aliquots (5 mL) of the culture medium were taken out, and centrifuged for 10 min at a speed of 10,000 rpm. The cell mass was separated through centrifugation using a centrifuge machine (SelectSpin Spectra 6C Shanghai, China). The supernatant was recovered via decantation. The supernatant absorbance was measured in the visible region of the UV–Visible spectrophotometer (UV-1800 ENG.SOFT) at the reported orange II absorption maximum wavelength (484 nm). The difference between the final and initial absorbance values was then used to compute the percentage degradation using the following formula [77].
(3)$$\%\;\mathrm{Degradation}=\frac{\mathrm{Initial}\;\mathrm{absorbance}-\;\mathrm{final}\;\mathrm{absorbance}}{\mathrm{Initial}\;\mathrm{absorbance}}\times 100$$

*B. subtilis* was chosen for a further dye decolorization (degradation) experiment because it had higher decolorization (85.55%) potential than of any other bacterial strains tested. 

### 3.6. Determination of Physiochemical Parameters for Optimization of Degradation Efficiency of B. subtilis

The proper degradation of dye by bacteria requires optimal conditions including the concentration of dye, time, temperature, pH, sodium chloride concentration, urea concentration, glucose concentration and redox mediators. The mentioned parameters were studied separately in each experiment. Each experiment, including the control, was repeated in triplicate and the mean values were considered. The details of these experiments are given below.

#### 3.6.1. Impact of Orange II Concentration on Degradation 

The *Bacillus subtilis* strain was cultured for 24 h in 8 test tubes containing 10 mL nutrient broth to investigate the impact of the concentration of dye on degradation/decolorization under static conditions. About 5 mL orange II solution from each concentration of the dye solutions (5, 10, 15, 20, 25, 30, 35 and 40 ppm, respectively) was added to different test tubes. A total of eight control solutions comprising 10 mL nutritional broth plus 5 mL orange 2 were made for each concentration of the dye. The initial absorbance values were recorded before decolorization. After a three-day incubation period, the degraded mixture was centrifuged at 10,000 rpm for 10 min in a centrifuge machine (SelectSpin Spectra 6C) at room temperature. The absorbance of the extracted supernatant was determined using a UV–Visible spectrophotometer (UV-1800 ENG.SOFT) [78].

#### 3.6.2. Impact of Time and Temperature on Orange II Degradation

The inoculated media plus the dye solution in the test tubes was incubated. Before decolorization, the initial absorbance was measured using a UV–Visible spectrophotometer. After a 1-day interval until 6 days, the absorbance of the isolated supernatant was again recorded. The percentage decolorization was then measured every 3 days until 21 days. 

To determine the effect of temperature, 5 mL orange II from the dye stock solution was added to the six test tubes containing 10 mL sterile nutrient broth inoculated with bacteria. Control or reference solutions containing 5 mL dye solution and 10 mL nutrient broth were also prepared. These tubes were then incubated at 25, 30, 35, 40, 45 and 50 °C in an incubator (Laboratory Hot Air Oven, DIGISystem Laboratory Instruments Corporation). The incubator was equipment to maintain the adjusted temperature constant. After a 3-day incubation period, the degraded sample was spun in the centrifuge at 10,000 rpm at room temperature for ten minutes. Using the UV–Visible spectrophotometer, the % decolorization/degradation was determined [79].

#### 3.6.3. pH Effect on Orange II Degradation

Proper pH is needed for the growth and survival of bacteria; therefore, the nutrient broth in the test tubes was inoculated with *B. subtilis* and incubated at 37 °C. Control solutions containing 10 mL broth plus 5 mL dye solution were also prepared. The pH values in the control solutions and in the inoculated test tubes were adjusted using 1M NaOH and 1M HCl solutions. By adding a minute quantity (1 microliter) of acid or base using a micropipette to the respective tubes, the pH values were recorded using pH indicator strips (Merck KGaA Darmstadt Germany). The initial absorbance values were recorded using the UV–Visible spectrophotometer. After 3 days, the mixtures in the test tubes were centrifuged in a centrifuge machine. The % degradation/decolorization of the supernatant was measured as described above [80]. 

#### 3.6.4. Impact of Glucose and Urea on Degradation of Orange II

A suitable amount of glucose is needed for bacteria as glucose is the main energy source for bacteria; additionally, it provides necessary carbon for bacterial biosynthesis. Sterile nutrient broth (10 mL) was added to the five test tubes and inoculated with selected bacterial culture. After visible bacterial growth and the addition of the dye solution, glucose at concentrations of 333.33 mg/L, 666.66 mg/L, 1000 mg/L, 1333.33 mg/L and 1666.67 mg/L was added to each test tube and incubated in an incubator at 37 °C. Control solutions containing 5 mL dye solution and 10 mL nutrient broth were also prepared for each concentration of glucose. 

A sufficient urea amount is required for the survival of bacteria, and also serves as a source of nitrogen. Amounts of 333.33 mg/L, 666.66 mg/L, 1000 mg/L, 1333.33 mg/L and 1666.67 mg/L were introduced to five inoculated test tubes containing dye solution. Control solutions of the dyes were also prepared. The percentage decolorization was determined at three-day intervals [31].

#### 3.6.5. Impact of Sodium Chloride Concentration on Orange II Degradation

The concentration of sodium chloride affects the dye degradation capability of bacteria because it is a major salt, and that is why it is present in high concentrations in water, coming out from sewers as waste into the water. The degradation of pollutants usually occurs at optimum saline conditions whereas high salt concentrations are intolerable for microorganisms. To five test tubes, we added 10 mL sterile broth nutrient and inoculated them with *B. subtilis* culture. The tubes were supplemented with 333.33 mg/L, 666.66 mg/L, 1000 mg/L, 1333.33 mg/L, and 1666.67 mg/L sodium chloride salt, respectively. The initial values of absorbance were measured. Control solutions were also prepared for each concentration. After centrifugation, the supernatant obtained was filtered and its percentage decolorization at the initial and final absorbance values recorded using Equation (3) [81].

#### 3.6.6. Redox Mediators’ Effect on Orange II Degradation

Bacteria use redox mediators during the electron transfer reaction, acting as electron acceptors and donors. These reagents mediate the oxidation reduction reaction during dye breakdown. In this study, hydroquinone, sodium benzoate, uric acid and ethylenediamine tetra acetic acid (EDTA) were used as redox mediators. Four test tubes containing 10 mL sterile nutrient broth inoculated with *B. subtilis* plus 5 mL dye solution were amended with uric acid, hydroquinone, ethylenediamine tetra acetic acid and sodium benzoate at a concentration of 66.66 mg/L. Four control solutions of the same concentrations of redox mediators were also prepared as references. The percentage decolorization was measured at 3-day intervals in a similar manner to that described above. It has been found that degradation experiments using the redox mediator can results in the highest decolorization rate [58].

### 3.7. Orange II Degradation at Optimum Conditions

In order to establish the ideal degradation, the optima of the tested parameters—dye concentration, pH, urea, temperature, sodium chloride, glucose, time and redox mediator concentration—were utilized and combined in a single final experiment. The percentage decolorization was estimated using the above procedure. 

All the experiments were performed in triplicate and the mean values were considered the outcome of the experiments. 

### 3.8. Extraction and Isolation of Metabolites after Orange II Biodegradation

The mixture obtained at optimum conditions containing the bacterial cell mass and degraded dye products was crushed and centrifuged for 20 min at 10,000 rpm in a centrifuge machine. For the metabolite’s extraction, the supernatant of the cell-free mass was used. Ethyl acetate was mixed with the supernatant in equal volume and shacked vigorously in a separating funnel for 30 min. The mixture was transferred to the separating funnel. In the separating funnel, organic and aqueous phases were separated from each other. The evaporation of ethyl acetate at 40 °C was ensured to obtain a solid crude extract of the mixture. 

#### 3.8.1. GCMS and FTIR Analysis of Orange II Metabolites

To identify the metabolites formed after degradation, an Agilent USB-393752 gas chromatograph (Agilent Technologies, Palo Alto, CA, USA) with an HHP-5MS 5% phenyl methyl siloxane capillary column (30 m × 0.25 mm × 0.25 μm film thickness; Restek, Bellefonte, PA, USA) equipped with an FID detector was used. The oven’s temperature was first held at 70 °C for one minute, then increased to 180° C for five minutes. The temperature of the machine was then raised to 280 °C for 20 min. The detector was then set to 290 °C, whereas the injector was heated to 220 °C. Helium as a carrier gas was used at a rate of 1 mL/min. One microliter of the sample was injected in split-less mode. The separated metabolites after GC were then examined by the GCMS system (Agilent HP-5973, Ramsey, Minneapolis, MN, USA) equipped with a mass-selective detector working in electron impact mode with an ionization energy of 70 eV [41]. The dye metabolites or degradation products were identified by comparing their retention times to those of the previously reported compounds in the literature. 

The FTIR analysis of dye and its degraded product mixture (metabolites) was performed using FTIR (PerkinElmer Spectrum Two instrument 103385; Waltham, MA, USA) [77]. 

#### 3.8.2. NMR Analysis of the Metabolites Formed after Biodegradation

The metabolites extracted in ethyl acetate were passed through a silica gel column for purification and isolation. Slurry was made by mixing the extract minute quantity with silica gel. The slurry was then subjected to evaporation to remove the solvent. A column was packed from silica gel to a height of 50 cm in a column 100 cm in height and 4 cm in diameter. The column was then washed with n-hexane for 2 h. The column was loaded with extract slurry at the top, washed with n-hexane, and then, eluted with ethyl acetate and n-hexane in different ratios. The fractions were collected from the silica gel column in small-sized glass vials with a 5 mL capacity. Similar isolates on the basis of TLC profiling and similar Rf values were combined. Metabolite confirmation was performed using thin-layer chromatography. Using TLC Profiling, only one fraction was identified. The purified fraction was then dried, dissolved in DMSO and subjected to FT-NMR (Advanced III 600 MHZ, Cryoprobe, Triple Resonance Chanel, Bruker, Billerica, MA, USA) for NMR spectroscopy.

### 3.9. Phytotoxicity Assay

The phytotoxicity experiment was conducted using a common agricultural plant *Vigna radiata* (green gram). Ten seeds were sown in separate Petri dishes and grown in an environment containing 25 ppm untreated and treated dyes (metabolites). The seeds were divided into Petri plates per the provided treatments: (i) control or tap-water treatment, (ii) untreated dye (orange II) treatment and iii) treated dye treatment. At regular intervals, 10 mL of tap water, and of untreated and treated dye water was added into each Petri plate. After 7 days, the percentage germination, and the length of shoot and root were measured. The % germination was calculated using the following formula or equation (4):(4)$$\%\;\mathrm{Germination}=\frac{\mathrm{Number}\;\mathrm{of}\;\mathrm{seeds}\;\mathrm{germinated}}{\mathrm{Total}\;\mathrm{number}\;\mathrm{of}\;\mathrm{seeds}}\times 100$$

### 3.10. Data Analysis

Throughout the study, all the experiments, including the control groups, were repeated in triplicate, and the results obtained are presented as the mean ± standard deviation. The optimization of degradation efficiency was investigated based on the interaction of process conditions using the central composite design (CCD) of the response surface methodology (RSM).

## 4. Conclusions

The present study aimed to evaluate the biodegradation potential of *B. subtilis.* The selected bacterial strain showed better degradation activity of the textile dye orange II. The effects of physicochemical conditions on bacteria’s degrading potential was determined. Finally, all of these physiochemical conditions were combined in a single final experiment. It can be determined from the result that the effect of the interactions of incubation time, pH, dye concentration and temperature enhanced the degradation of the bacterial strain at a desirability coefficient of 1 and at a correlation coefficient (R^2^) value of 0.8950. The metabolites formed at optimum conditions were analyzed using FTIR, GCMS and NMR spectroscopic techniques. The result of the characterization also revealed that the dye was degraded by the bacteria, and through subsequent enzymatic action, naphthalene and o-xylene were formed. Naphthalene was formed through orange II dye azo bond reduction by the bacterial enzyme azo reductase, followed by deamination and a reduction in the 1-amino 2-napthol ring. Another metabolite, o-xylene, was formed by the desulfonation and subsequent deamination of sodium-4-aminobenzene sulphonate, followed by the methylation of the formed benzene molecule. Moreover, the phytotoxicity results described that dye metabolites were less toxic than the parent dye (orange II). From the above results we conclude that *B. subtilis* could be used as an effective strain for the treatment of textile dyes present in wastewater. Biodegradation of the azo dye orange II using *B. subtilis* in aerobic conditions is a green and ecofriendly biotechnological approach to the detoxification of the dyes in textile effluents; however, further research is needed in this area to isolate the enzymes from the selected bacteria. More labor is needed to study the biodegradation potential of *Bacillus subtilis* for other azo dyes. Functional genes of this bacterium should be transferred to others, and genetically modified bacteria should be used for better results in this field. The desired enzymes from *B. subtilis* should be isolated using advanced techniques in biotechnology and molecular biology. Other optimum physiochemical and environmental conditions necessary for this bacterium should be determined for better and optimum degradation. The effectiveness of *B. subtilis* for the degradation of orange II dye indicates its potential and its future application in advanced bioreactor systems in the industrial sector for the treatment of textile effluents and wastewater before their release into the environment.

## Figures and Tables

**Figure 1 ijms-23-10637-f001:**
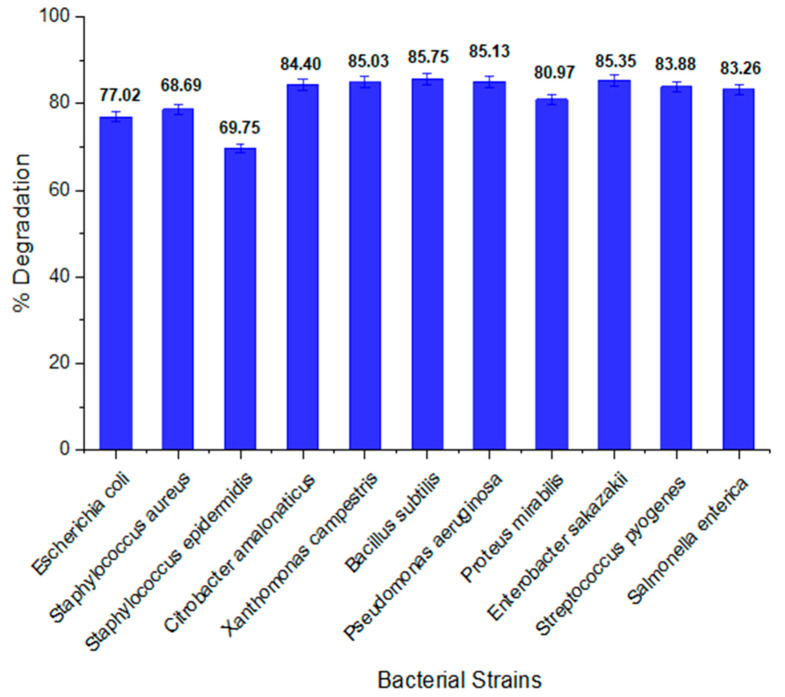
Percentage degradation of orange II by bacterial strains.

**Figure 2 ijms-23-10637-f002:**
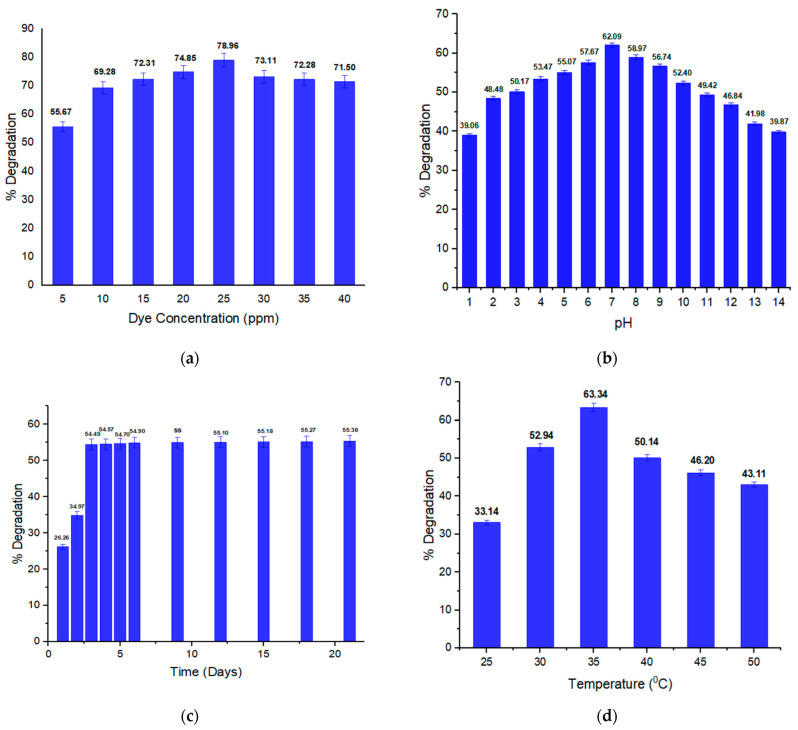
Effects of (**a**) dye concentration, (**b**) pH, (**c**) incubation duration (time), (**d**) temperature, (**e**) glucose concentration, (**f**) urea concentration, (**g**) sodium chloride concentration and (**h**) redox mediators on degradation of orange II.

**Figure 3 ijms-23-10637-f003:**
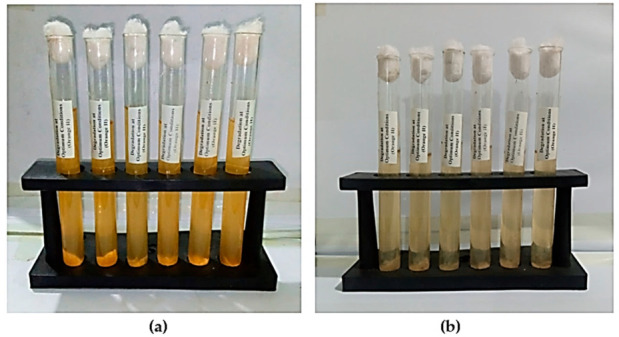
Orange II dye (**a**) before bacterial degradation and (**b**) after *B. subtilis* treatment.

**Figure 4 ijms-23-10637-f004:**
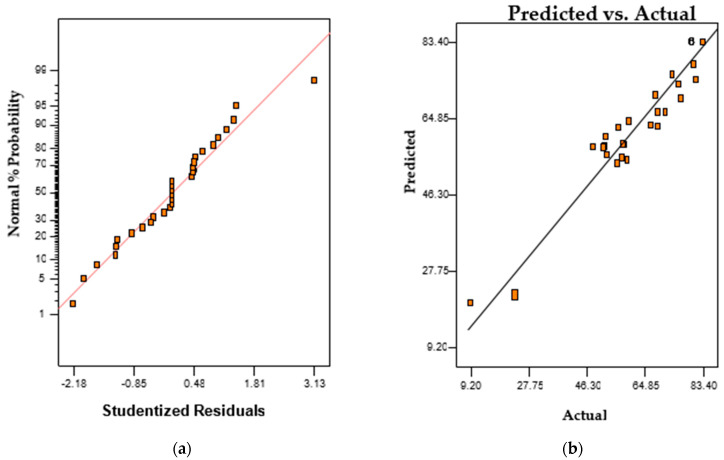
Relationship between outcomes: (**a**) probability coefficient and (**b**) predicted and actual.

**Figure 5 ijms-23-10637-f005:**
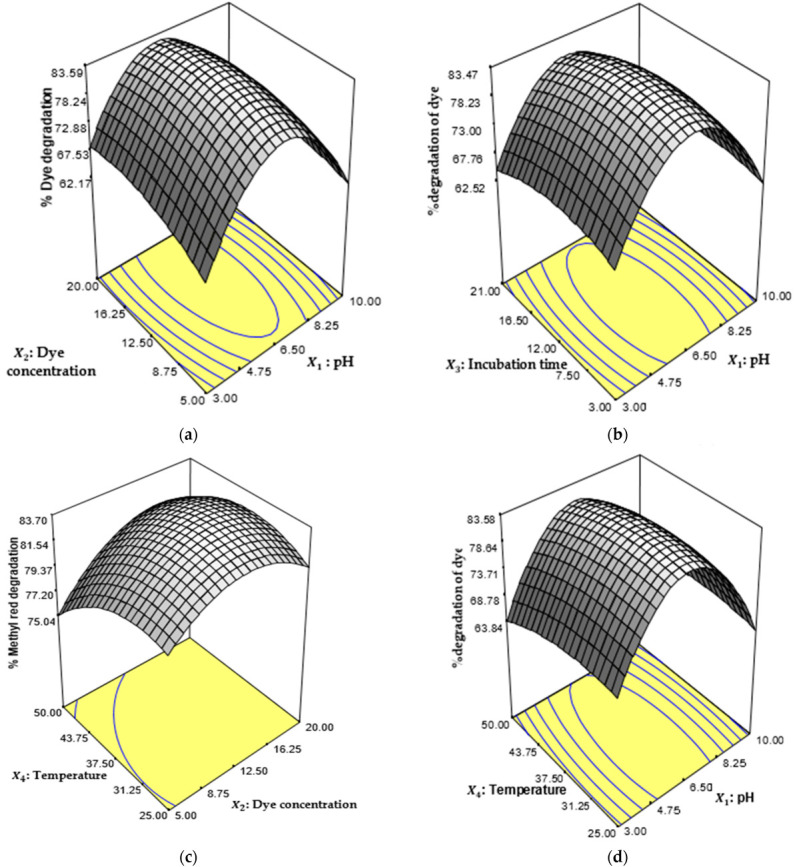
Three-dimensional surface plot for the interaction of (**a**) pH and dye concentration, (**b**) pH and incubation time, (**c**) dye concentration and temperature and (**d**) pH and temperature for the biodegradation of orange II.

**Figure 6 ijms-23-10637-f006:**
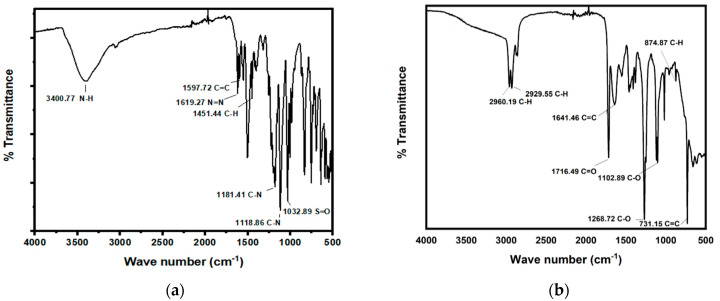
(**a**) FTIR spectra before degradation of orange II and (**b**) FTIR spectra after orange II degradation by *B. subtilis*.

**Figure 7 ijms-23-10637-f007:**
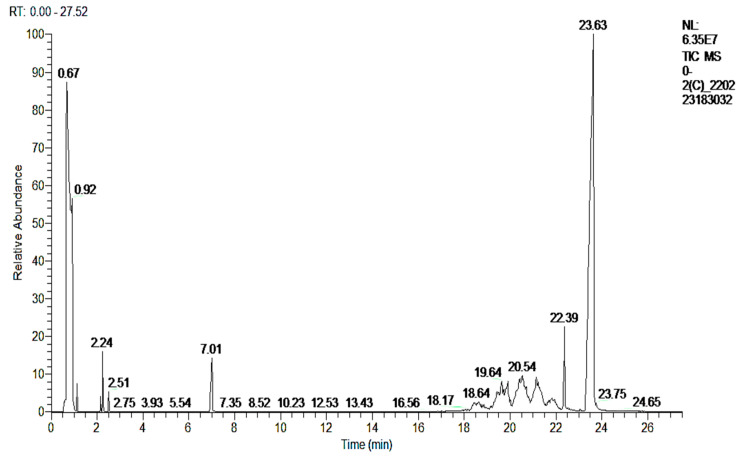
GC chromatogram of orange II after *B. subtilis* degradation.

**Figure 8 ijms-23-10637-f008:**
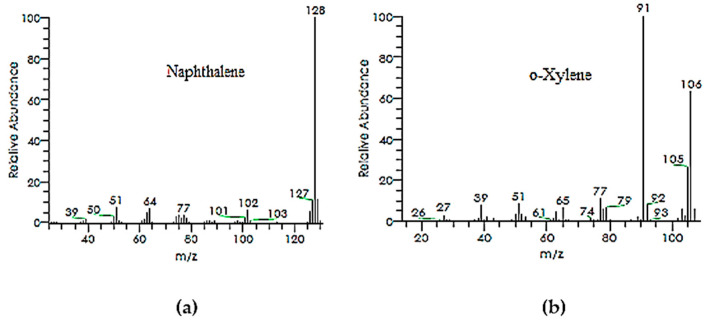
(**a**,**b**) GCMS chromatograms of two important metabolites.

**Figure 9 ijms-23-10637-f009:**
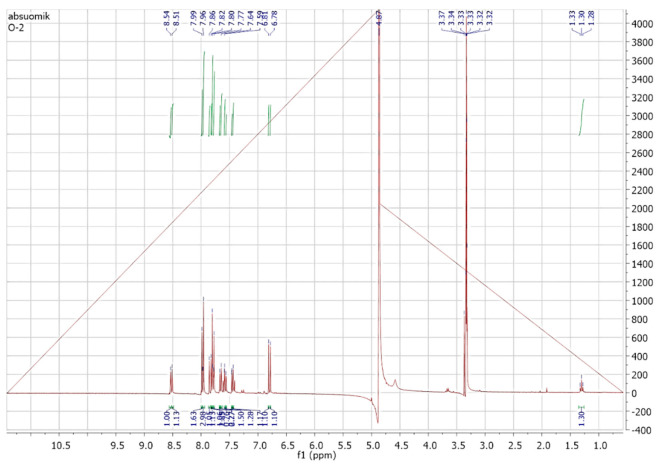
^1^H NMR of the original dye, orange II.

**Figure 10 ijms-23-10637-f010:**
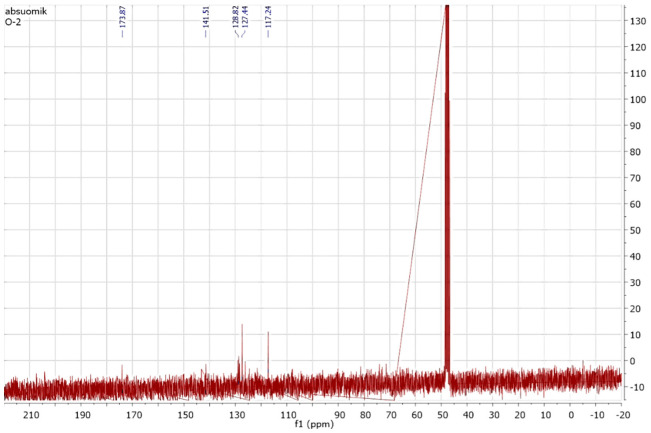
^13^C NMR of the original dye, orange II.

**Figure 11 ijms-23-10637-f011:**
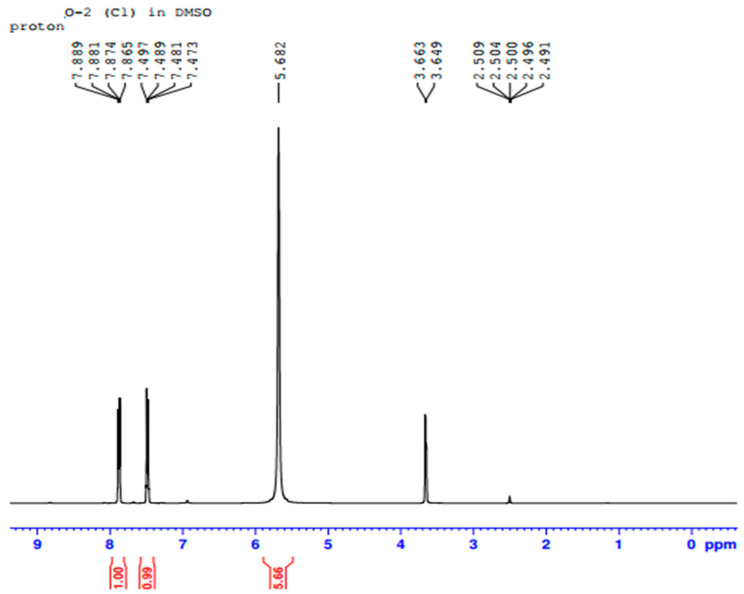
^1^H NMR of isolated metabolite.

**Figure 12 ijms-23-10637-f012:**
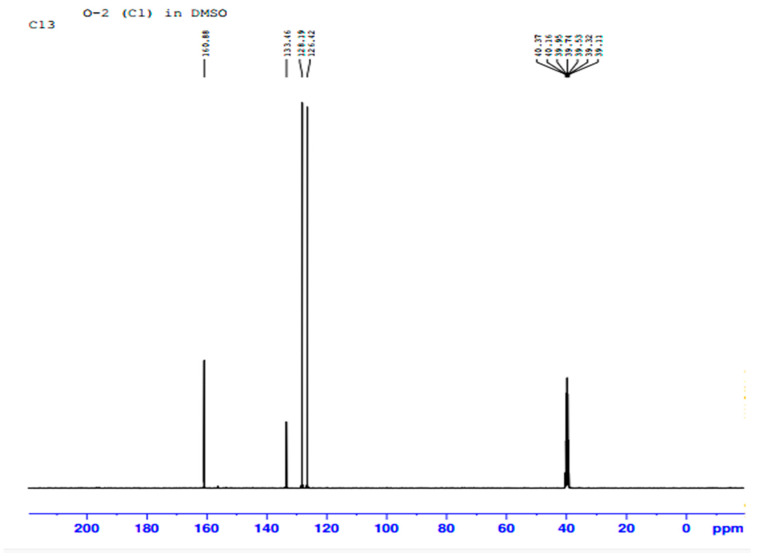
^13^C NMR of isolated metabolite.

**Figure 13 ijms-23-10637-f013:**
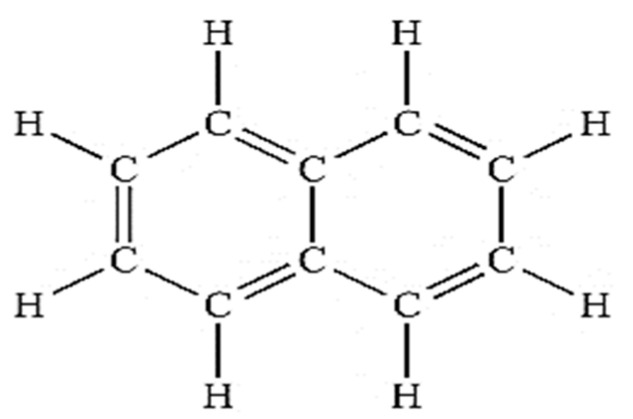
Chemical structure of metabolite (naphthalene).

**Figure 14 ijms-23-10637-f014:**
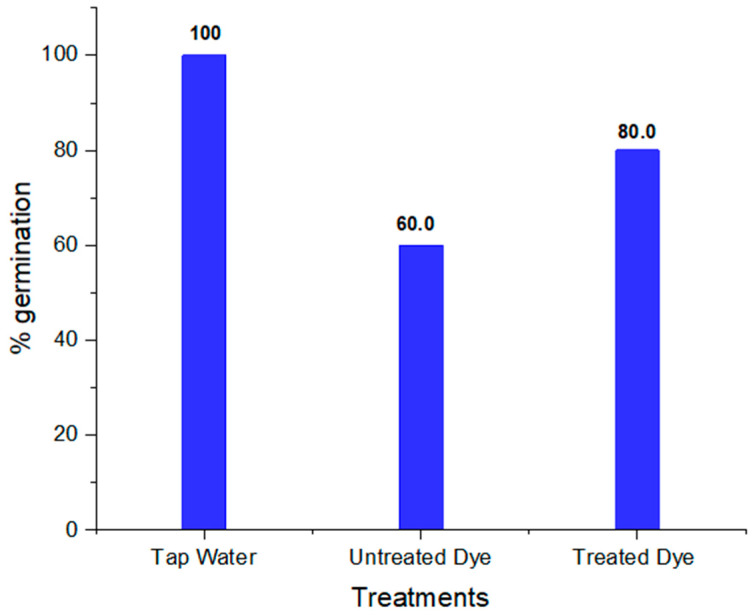
Percentage germination of *Vigna radiata* seeds in tap water and in untreated and treated orange II dye.

**Figure 15 ijms-23-10637-f015:**
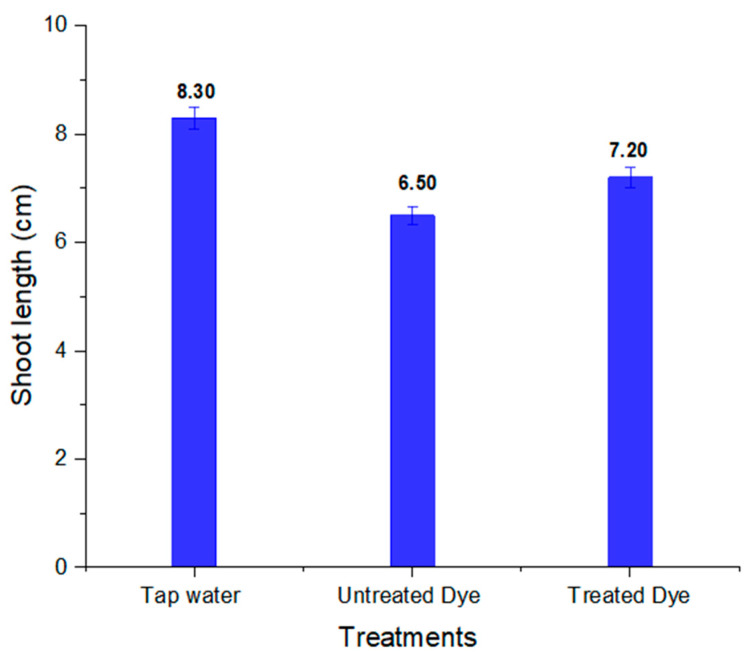
Shoot length of *Vigna radiata* in tap water and in untreated and treated orange II dye.

**Figure 16 ijms-23-10637-f016:**
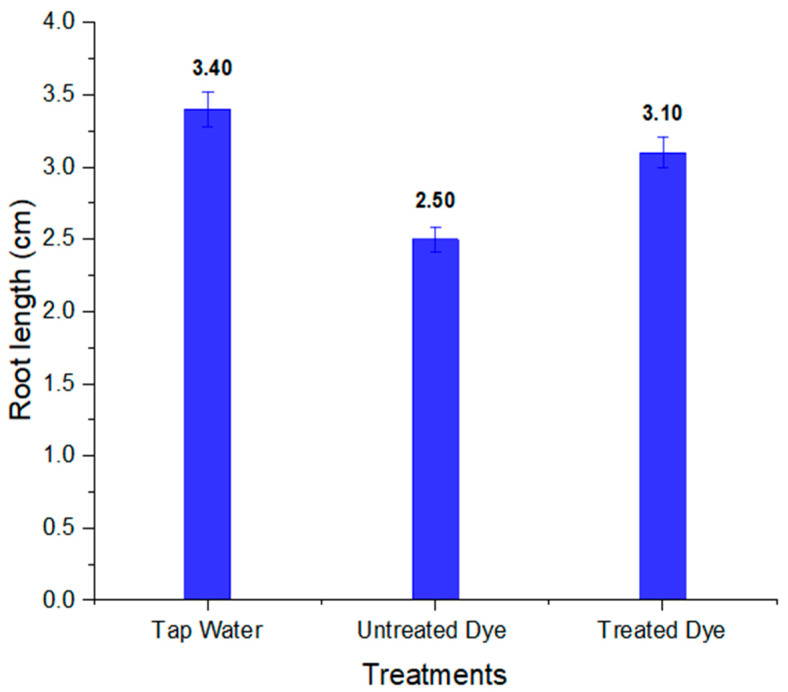
Root length of *Vigna radiata* in tap water and in untreated and treated orange II dye.

**Figure 17 ijms-23-10637-f017:**
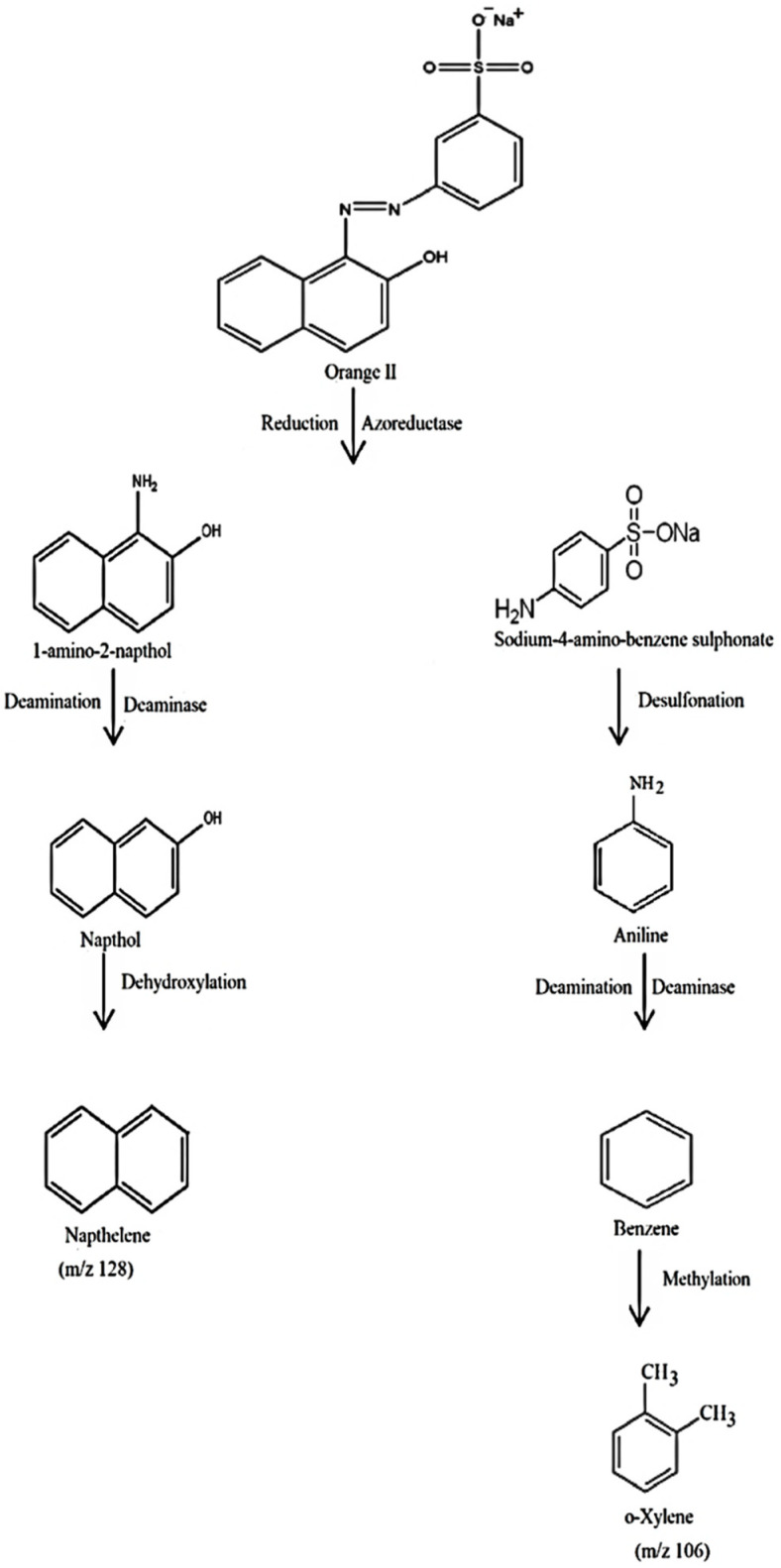
Proposed mechanism for orange II dye degradation by *B. subtilis*.

**Figure 18 ijms-23-10637-f018:**
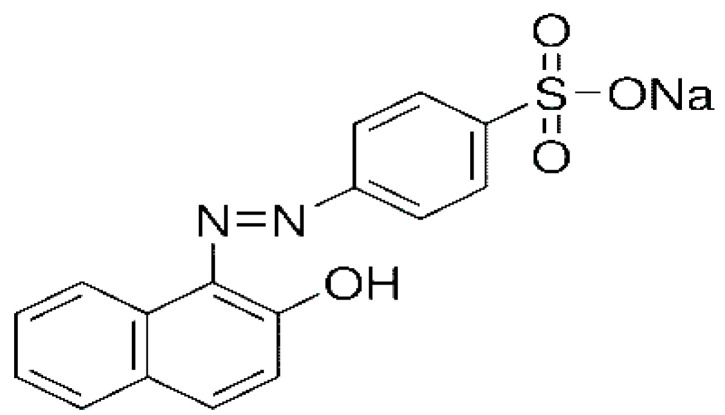
Orange II chemical structure.

**Table 1 ijms-23-10637-t001:** Central composite design for the degradation of orange II dye.

Factors	Units	Code	Levels		
			−1	0	+1
pH	-	$X_{1}$	3	6.5	10
Dye concentration	ppm	$X_{2}$	5	22.5	40
Incubation time	day	$X_{3}$	3	21	12
Temperature	°C	$X_{4}$	25	37.5	50

**Table 2 ijms-23-10637-t002:** Central composite design matrix.

Run	$X_{1}$	$X_{2}$	$X_{3}$	$X_{4}$	Observed Value	Predicted Value
1	0	0	2	0	47.61	36.95
2	−1	−1	−1	1	37.20	68.30
3	1	−1	1	1	31.40	56.10
4	0	−2	0	0	37.95	76.40
5	−1	−1	−1	−1	56.60	52.80
6	−1	1	1	−1	58.20	68.90
7	1	1	1	−1	57.89	52.70
8	0	0	0	2	64.20	75.60
9	−1	1	−1	−1	58.43	71.40
10	0	0	0	0	75.40	83.40
11	0	0	0	0	68.30	83.40
12	0	2	0	0	68.20	73.60
13	−1	1	1	1	70.12	66.60
14	1	−1	1	−1	72.90	80.30
15	−1	−1	−1	1	67.80	81.20
16	0	0	0	0	63.21	83.40
17	1	−1	1	−1	42.30	56.40
18	−1	1	−1	1	61.80	68.90
19	−2	0	0	0	72.15	19.2
20	−1	1	1	1	81.20	57.80
21	1	−1	−1	−1	71.23	58.20
22	1	1	−1	−1	69.20	52.30
23	1	1	−1	1	61.20	59.70
24	1	−1	−1	1	68.93	48.30
25	0	0	0	0	69.87	83.40
26	1	1	1	1	69.20	32.60
27	0	0	0	0	65.80	83.40
28	2	0	0	0	69.73	61.70
29	−1	−1	1	1	59.87	59.20
30	0	0	0	0	69.20	83.40

**Table 3 ijms-23-10637-t003:** ANOVA percentage degradation of orange II dye.

Source	Sum of Squares	DF	Square Values	*F*-Value	*p*-Value
Model	7509.07	14	536.36	9.13	<0.0001
$X_{1}$	12.61	1	12.61	0.21	0.0498
$X_{2}$	51.63	1	51.63	0.88	0.0363
$X_{1}{}^{2}$	434.09	1	434.09	8.77	0.0097
$X_{3}{}^{2}$	207.74	1	207.74	7.59	0.0147
$X_{1}X_{2}$	36.69	1	36.69	3.67	0.0074
Lack of fit	-	-	-	-	0.6246
Mean	PRESS	Adeq Precision	R-Squared	Adj R-Squared	Std. Dev.
64.51	5076.88	12.233	0.89.50	0.7969	7.67

**Table 4 ijms-23-10637-t004:** Identified metabolites in degraded dye mixture from GCMS results.

S. No	Metabolite	Peak Area	Retention Time	Chemical Formula	Molecular Weight
1.	Naphthalene	2.29	6.99	C_10_H_8_	128
2.	o-Xylene	1.42	2.24	C_8_H_10_	106

**Table 5 ijms-23-10637-t005:** Phytotoxicity of orange II dye and its metabolites in *Vigna radiata*.

S.No	Parameters	Tap Water	Dye	Treated Dye
1	% germination	100	60	80
2	Shoot length (cm)	8.30 ± 0.20	6.50 ± 0.16	7.20 ± 0.18
3	Root length (cm)	3.40 ± 0.11	2.50± 0.08	3.10 ± 0.10

## Data Availability

Not applicable.

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
