# Peer review of "Bacillus subtilis*: As an Efficient Bacterial Strain for the Reclamation of Water Loaded with Textile Azo Dye, Orange II"

_ijms, 2022, doi:10.3390/ijms231810637_

Round 1

Reviewer 1 Report

1. The manuscript needs to be modified carefully on writing. There are many formatting, grammatical and spelling errors throughout the manuscript.

2. Latin name (Bacillus subtilis) should be spelled out when it first appear, then abbreviated (B. subtilis). Please check the text and correct the format.

3. Names of bacterial strains are always written in italics. Please check the text and make correction.

4. Check the citation of references.

Author Response

Reviewer 1 Comment:

The manuscript needs to be modified carefully on writing. There are many formatting, grammatical and spelling errors throughout the manuscript. Response: The manuscript has been revised accordingly by a native speaker. Thank you Comment: Latin name (Bacillus subtilis) should be spelled out when it first appears, then abbreviated (B. subtilis). Please check the text and correct the format. Response: This mistake has been corrected throughout manuscript. Comment: Names of bacterial strains are always written in italics. Please check the text and make correction. Response: These mistakes has been corrected throughout manuscript. Thank you. Comment: Check the citation of references. Response: The citation of references and format has been checked and mistakes has been corrected.

Reviewer 2 Report

The research paper reports a very interesting approach as although dyes decolorization by microorganism has been reported by many researchers but no one has successfully isolated the metabolites and have elaborated the mechanism of degradation. The work is valuable and could accepted after some minor corrections:

1.      Revise the keyword list and arrange them alphabetically 

2.      In introduction add some details about the bacterial strain used. Also point out whether it is beneficial or hazardous bacteria as if researcher use it in future must then follow precautions if hazardous.

3.      In introduction write the synonyms of dye in brackets as they sometime creates confusion in readers minds

4.      In equations the text format should be uniform

5.      Why glucose, NaCl and Urea has been added to growth cultures

6.      There are minor spelling mistakes please correct them. 

7.      The references style is not uniform

Author Response

Reviewer 2

Comment: The research paper reports a very interesting approach as although dyes decolorization by microorganism has been reported by many researchers but no one has successfully isolated the metabolites and have elaborated the mechanism of degradation. The work is valuable and could accepted after some minor corrections:

Response: Worthy reviewer thank you very much for your appreciation of our research work 

 Comment: Revise the keyword list and arrange them alphabetically 

Response: The key words has been revised and arranged alphabetically.

Comment:  In introduction add some details about the bacterial strain used. Also point out whether it is beneficial or hazardous bacteria as if researcher use it in future must then follow precautions if hazardous.

Response: The detail about bacterial strain has been added in introduction section accordingly. Thanks.

Comment: In introduction write the synonyms of dye in brackets as they sometime create confusion in readers minds

Response: The synonym of dye used in the study has been added along with dye in brackets.

Comment:  In equations the text format should be uniform.

Response: The text format has been uniformly revised.

Comment:  Why glucose, NaCl and Urea has been added to growth cultures.

Response: Suitable amount of glucose is needed for bacteria as glucose is the main energy source for bacteria; also, it provides carbon necessary for bacteria to degrade the dye effectively. Degradation of pollutants usually occurs at optimum saline conditions as it balance the osmotic pressure of bacterial system. Urea serves as additional source of nitrogen for protein synthesis of bacteria as a result increases dye degradation potential of bacteria. Addition of nitrogen can also regenerate coenzyme NADH, that acts as electron donor for the reduction of azo dyes. Degradation of these dyes is difficult without the supplementation of additional sources of carbon and nitrogen and salts in appropriate amount/concentration is needed.

Comment:  There are minor spelling mistakes please correct them. 

Response: The spelling mistakes has been corrected. Thank you

Comment:  The references style is not uniform

Response: The references styles has been checked and revised. Thank you